# Sex- and Gender-Specific Considerations in Mycotoxin Screening: Assessing Differential Exposure, Health Impacts, and Mitigation Strategies

Gayathree Thenuwara [1], Bilal Javed [1,2], Baljit Singh [2,3], Hugh J. Byrne [2] and Furong Tian [1,2,*]

1    School of Food Science and Environmental Health, Technological University Dublin, Grangegorman, D07 ADY7 Dublin, Ireland; gayathreethenuwara57@gmail.com (G.T.); bilal.javed@tudublin.ie (B.J.)
2    Nanolab Research Centre, Physical to Life Sciences Research Hub, Technological University Dublin, Camden Row, D08 CKP1 Dublin, Ireland; baljit.singh@tudublin.ie (B.S.); hugh.byrne@tudublin.ie (H.J.B.)
3    MiCRA Biodiagnostics Technology Gateway, Technological University Dublin (TU Dublin), D24 FKT9 Dublin, Ireland
*    Correspondence: furong.tian@tudublin.ie; Tel.: +353-1220-5673

**Abstract:** Mycotoxins, toxic secondary metabolites produced by fungi, present significant health risks through contaminated food and feed. Despite broad documentation of their general impacts, emerging research highlights the requirement of addressing both sex- and gender-specific differences in the risk of exposure, susceptibility, and health outcomes in mycotoxin screening and mitigation strategies. Distinct biological (sex-based) and sociocultural (gender-based) factors can influence the risk of mycotoxin exposure and subsequent health impacts; women may for example exhibit specific exposures to certain mycotoxins due to physiological and hormonal differences, with increased risks during critical life stages such as pregnancy and lactation. Conversely, men may demonstrate distinct metabolic and immune responses to these toxins. Socioeconomic and cultural factors also contribute to gender-specific exposure risks, including occupational exposures, dietary habits, and healthcare access. Current mycotoxin screening methodologies and regulatory frameworks often disregard these sex and gender disparities, resulting in incomplete risk assessments and suboptimal public health interventions. This review addresses the incorporation of sex- and gender-specific data into mycotoxin research, the development of advanced screening techniques, and the implementation of targeted mitigation strategies. Addressing these sex and gender differences is crucial for enhancing the efficacy of mycotoxin management policies and safeguarding public health. Future research directions and policy recommendations are discussed to promote a more comprehensive and practical approach to mycotoxin risk assessment and control.

**Keywords:** mycotoxins; sex and gender differences; exposure pathways; health impacts; physiological vulnerability; metabolic responses; immune reactions; risk assessment; mitigation strategies

## 1. Introduction

Mycotoxins, toxic secondary metabolites produced by fungi, present a substantial global challenge to food safety and environmental health due to their high toxicity and widespread occurrence [1]. These contaminants impact a wide range of agricultural products, including grains, nuts, spices, and fruits, posing significant risks to both humans and animals even at low concentrations [1]. Over 500 mycotoxins have been documented to date, and ongoing research is suggesting that this inventory is not yet complete [2,3]. Commonly encountered mycotoxins include aflatoxins (AFs), ochratoxin A (OTA), fumonisins, trichothecenes (DON), and zearalenone (ZEN) [1–3].

Fungi from genera such as *Aspergillus*, *Alternaria*, *Fusarium*, *Penicillium*, *Claviceps*, and *Stachybotrys* are primary producers of these mycotoxins [1–3]. These fungi can contaminate crops under specific environmental conditions, such as temperature and humidity, leading

to mycotoxin accumulation before harvest and during storage. Despite advancements in agricultural practices, mycotoxin contamination remains prevalent, with approximately 70% of raw materials being affected [4]. The Food and Agriculture Organization (FAO) of the United Nations estimates that around 25% of global food crop production is impacted by mycotoxin contamination [5].

Assessing the risks associated with mycotoxin contamination in food involves considering several complex factors. One significant challenge is the presence of emerging mycotoxins, for which limited toxicological data are available [6,7]. While advances in detection technologies continually identify new mycotoxins, their risks remain poorly defined until comprehensive toxicological studies are conducted [6,7]. Another critical factor is the occurrence of mycotoxins in modified forms, which can be more toxic than their parent compounds [8–10]. These modifications can arise in plants, as a defense mechanism, or during food and feed processing [8–10]. Additionally, the interaction between multiple mycotoxins presents a further layer of complexity [11,12]. Mycotoxins often co-occur in food and feed due to the simultaneous presence of multiple fungal species or the contamination of various commodities. The combined effects of these mycotoxins can be antagonistic, additive, or synergistic, leading to toxic outcomes that are not predictable by evaluating each toxin individually [11,12]. Thus, even when mycotoxins are present at levels considered safe on their own, their combined presence can still pose significant health risks [1,13].

The diverse toxic effects of mycotoxins are linked to their ability to induce a range of health issues, including carcinogenicity, nephrotoxicity, hepatotoxicity, estrogenicity, neurotoxicity, and alterations to reproductive and immune systems [14]. For instance, aflatoxins, particularly aflatoxin B1 (AFB1), predominantly target the liver, causing toxicity through mechanisms such as the disruption of protein synthesis, oxidative stress, and cellular damage [14]. Similarly, fumonisins, notably fumonisin B1 (FB1), are associated with several adverse health outcomes, including neural tube defects, embryonic and fetal toxicity, and impaired growth in children [15,16].

In addition to aflatoxins and fumonisins, trichothecenes, such as deoxynivalenol (DON), primarily affect rapidly proliferating tissues, including those in the hematopoietic, lymphoid, and gastrointestinal systems. This results in symptoms such as abdominal pain, vomiting, diarrhea, and growth retardation [17]. Ochratoxin A (OTA) stands out as a potent nephrotoxin, with primary effects on the kidneys and links to conditions such as Balkan endemic nephropathy (BEN), renal failure, and renal cancer [18]. Patulin (PAT) is associated with gastrointestinal disturbances [14], while zearalenone (ZEN) causes reproductive disorders and precocious puberty [19].

Host-related factors significantly influence the in vivo effects of mycotoxins, with sensitivity varying based on differences in digestive physiology, metabolism, excretion capabilities, and anatomical features. These factors, which include species, sex, age, nutritional status, pre-existing diseases and microbiota, play crucial roles in determining the onset and severity of mycotoxin exposure effects [19–25]. Among these, sex stands out as a particularly influential factor, impacting the response to mycotoxins due to hormonal differences and variations in pharmacokinetics and pharmacodynamics [26–29]. Hormonal influences on hepatic detoxifying enzyme expression, along with intrinsic differences in cell composition and structure, contribute to significant sexual dimorphism in toxic responses [26–29].

Research has documented notable differences in how males and females respond to mycotoxins. A meta-analysis of studies on pigs, covering 85 articles from 1968 to 2010, found that mycotoxin exposure generally had a more pronounced impact on males [30]. Specifically, the feed intake was reduced by 10% in males compared to 6% in females, weight gain was diminished by 19% in males versus 15% in females, and the feed conversion ratio worsened by 10% in males compared to 8% in females. Despite these findings, sex differences have often been overlooked in mycotoxicological research, with females frequently being under-represented. A 2015 review of in vivo toxicity studies on *Fusarium*

*mycotoxins* revealed that 54% of studies used only males, 15% used only females, 15% included both sexes, and 15% did not specify the sex [31]. This lack of consideration for sex differences can lead to erroneous conclusions when extrapolating data across sexes. Males are often preferred in research due to their more stable hormone levels, which result in less variability. Figure 1 presents a comprehensive analysis of the impact of mycotoxins on both male and female reproductive health, as well as pregnancy outcomes. The figure illustrates how exposure to various mycotoxins, such as ZEN, Alternariol, AFB1, OTA, fumonisin, and PAT, disrupts hormonal balance, impairs fertility, and negatively affects reproductive functions in both genders.

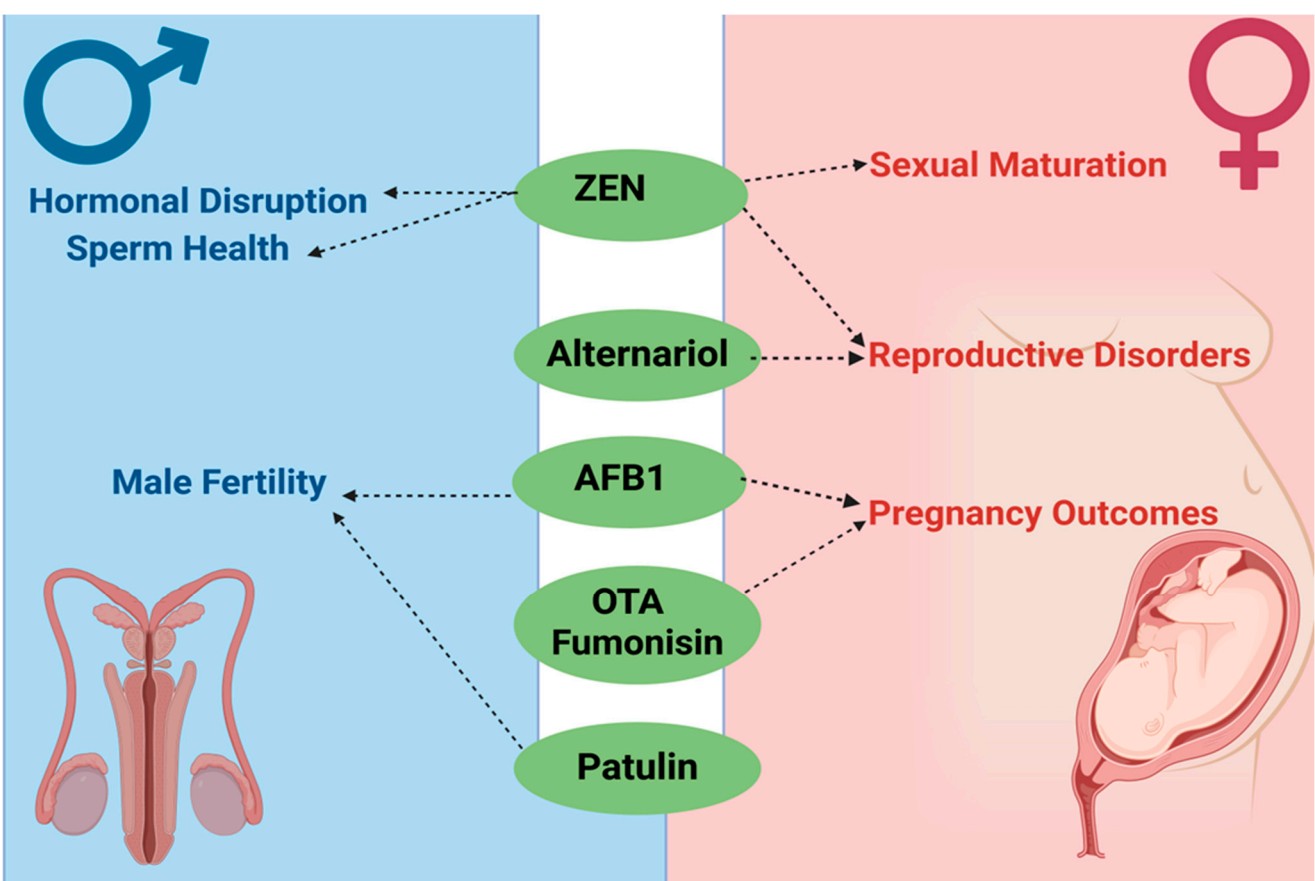

**Figure 1.** Sex-specific reproductive toxicity of mycotoxins: impact on male hormonal balance, sperm health, and fertility, and female sexual maturation, reproductive disorders, and pregnancy outcomes (created with BioRender).

For females, these toxins are shown to affect sexual maturation, cause reproductive disorders, and impair pregnancy outcomes by disrupting hormonal regulation and causing developmental issues in the fetus. Meanwhile, in males, the figure highlights the impact on sperm quality, hormonal disruption, and reduced fertility. Together, this figure provides an integrated view of how mycotoxins can influence reproductive health in sex-specific ways, emphasizing the critical risks posed by these environmental contaminants to overall reproductive function.

In addition to physiological and hormonal differences, occupational exposures also play a significant role in gender-specific mycotoxin exposure. Men and women often engage in different types of work, resulting in varied exposure levels. For instance, men working in agriculture or food processing may encounter mycotoxins directly through contaminated crops and dust, while women may experience exposure through handling contaminated food products in domestic or food preparation settings. These occupational

differences complicate the understanding of mycotoxin exposure and highlight the need for gender-tailored risk assessments and mitigation strategies.

The review by Marcelloni et al. (2024) underscores the diverse occupational exposure to mycotoxins across various industries [32]. Analyzing 31 articles from 2010 to 2024, the review reveals that mycotoxins, typically associated with alimentary exposure, also pose significant health risks through inhalation in occupational settings [32]. Mycotoxins can become airborne, attaching to dust particles and fungal spores, exposing workers in food and feed industries, silos, warehouses, grain transport, waste treatment, and agricultural machinery maintenance [32]. In particular, animal feed processing plants present higher mycotoxin levels, increasing risk for workers. The review highlights that sectors such as agriculture, animal care, waste treatment, healthcare, and others face different exposure levels [32]. For example, feed handling in animal husbandry and grain processing in agriculture are major exposure sources. Healthcare settings generally show lower exposure, though heating, ventilation, and air conditioning systems can be contamination sources. Waste treatment plants also present risks during waste handling, and even libraries and archives can pose risks due to fungal contamination [32].

Similarly, the 2018 review by Viegas et al. thoroughly evaluated mycotoxin exposure across various occupational settings, including agriculture, food processing, and animal husbandry [33]. Analyzing 15 studies published between 1981 and 2017, the review identified significant levels of mycotoxins such as OTA and aflatoxins in environments like grain elevators, poultry farms, and food processing facilities. The study found elevated concentrations of these mycotoxins in airborne dust and surface samples, underscoring substantial health risks to workers. Both reviews collectively highlight the critical need for improved monitoring and control measures to protect workers' health and reduce the risk of mycotoxin-related diseases across various occupational settings [33].

Despite the extensive research on occupational exposure to mycotoxins, there is a notable scarcity of studies that comparatively analyze exposure patterns with respect to gender. Although occupational roles and exposure levels may differ significantly between men and women due to varying job functions and environments, research specifically addressing these gender-based differences remains limited. This gap in the literature highlights the urgent need for more focused studies to elucidate how gender influences mycotoxin exposure and to inform the development of gender-specific risk assessments and mitigation strategies.

Males and females tend to consume different types and quantities of food, leading to varying levels of mycotoxin exposure [34]. Staples such as corn, wheat, and rice are common sources of mycotoxins like aflatoxins, ochratoxin A, and deoxynivalenol [35].

Studies have shown that males often consume larger quantities of these staples compared to females, which can be influenced by their higher body mass and caloric requirements, potentially increasing their exposure risk to mycotoxins [34]. Foods like peanuts and tree nuts, which are susceptible to aflatoxin contamination, are also consumed in varying amounts between genders, with males typically having higher intake levels, contributing to a greater risk of exposure [34]. Although fruits and vegetables can harbor mycotoxins such as patulin and ochratoxin A, consumption patterns may be more balanced or even higher among females due to health-conscious dietary choices [34].

Cultural dietary practices play a significant role in shaping food consumption patterns, impacting respective mycotoxin exposure levels. In certain cultures, traditional diets dictate specific food consumption for males and females, influencing their respective mycotoxin exposure levels [36,37]. For instance, in some regions, men may have access to more diverse food options, while women may rely more heavily on staple grains. Socioeconomic status also affects dietary patterns and exposure to contaminated food. In lower socioeconomic groups, both males and females might consume more affordable, potentially contaminated food staples, but the quantity and frequency can differ between genders [38].

Gender-specific elements in mycotoxin screening may significantly improve detection and management approaches. By adjusting screening techniques to consider the unique

responses of males and females to mycotoxin exposure, sensitivity and safety thresholds can be optimized. Targeted investigations involving both males and females are essential to validate screening methods across genders and identify biomarkers or changes in mycotoxin metabolism. Customized screening tests for specific occupational environments, such as agriculture, food processing, and healthcare, can provide more precise evaluations. Gender-specific risk modeling should consider variations in occupational exposure and physiological reactions, resulting in more accurate risk evaluations and improved screening methods. Advancements in screening methods, such as the identification of biomarkers specific to gender and the integration of digital tools, can significantly enhance detection accuracy. Integrating gender-specific data into point-of-care testing can improve detection capacities and dependability. This strategy can lead to improved public health results and safer interventions to reduce risks linked to mycotoxin contamination.

## 2. Sex Differences in Mycotoxin Exposure

### 2.1. Sex-Specific Effects of Mycotoxins on Reproductive Health, Immune Responses, Cancer Risk, and Pregnancy Outcomes

2.1.1. Impact of Mycotoxins on the Human Reproductive System

Exposure to mycotoxins throughout an individual's life, from fetal development to adulthood, can significantly affect various body systems, primarily due to alterations in hormonal balance caused by endocrine-disrupting chemicals (EDCs) [39,40]. EDCs are known to interfere with growth-promoting hormones, such as insulin and insulin-like growth factors 1 and 2, while also enhancing the effects of glucocorticoids, which can inhibit these growth-promoting processes [41]. For instance, ZEN, an EDC, can disrupt endocrine functions in the placenta and kidneys by selectively inhibiting 11β-hydroxysteroid dehydrogenase type 2 (HSD11B2) [42]. This disruption might also stem from ZEN's pro-inflammatory effects or its interaction with estrogen receptors [43]. Hormones like growth hormones from the pituitary gland and thyroid hormones can be adversely affected by EDCs, potentially leading to developmental disorders such as growth abnormalities and delayed puberty [44,45].

The potential impact of mycotoxin exposure on sexual development and reproductive health is severe; therefore, comprehensive analysis should be emphasized. Numerous epidemiological studies have demonstrated a significant impact of ZEN on the age of pubertal maturation, resulting in premature sexual development particularly of girls with rising levels of estrogen [46–49]. Similar findings were reported by Yum et al., who observed premature puberty in boys under the age of nine [45]. Elevated ZEN levels (ranging from 18.9 to 103 μg/L) were also documented in girls with an early onset of breast development and precocious puberty in studies conducted in Turkey, Hungary, and Italy [46,50,51]. Bandera et al. reported accelerated breast development in girls aged 9–10 years [48], while Rivera-Núñez found that girls with detectable mycoestrogen levels had lower rates of sexual maturation compared to their peers without detectable levels [49]. ZEN also adversely affects reproductive health by disrupting the production and secretion of key sex hormones, such as estradiol, progesterone, and testosterone, potentially leading to infertility [52].

Despite limited research, there is a consensus that mycoestrogens significantly disrupt ovarian folliculogenesis [53,54]. Mycotoxins are implicated in conditions such as polycystic ovary syndrome (PCOS), premature ovarian failure (POF), and endometriosis, all of which can complicate conception and pregnancy maintenance [55]. In males, ZEN impairs fertility by binding to estrogen receptors, which increases estradiol levels and inhibits luteinizing hormone (LH) synthesis in the pituitary gland. This results in reduced testosterone production in Leydig cells, which are located in the interstitial tissue of the testes, thereby affecting spermatogenesis. Additionally, ZEN decreases sperm motility, viability, and acrosomal response, and induces the formation of reactive oxygen species, further compromising spermatogenesis [56].

AFB1 is another mycotoxin suspected to impact male fertility. A study conducted in Nigeria found higher concentrations of AFB1 (60 ng/mL to 148 ng/mL) in the semen of infertile men compared to fertile men (0 ng/mL to 5 ng/mL) [57]. This study also reported that fertility disorders were associated with elevated AFB1 levels exceeding World Health Organization (WHO) standards [58].

The evidence suggests that exposure to mycotoxins, particularly ZEN and AFB1, is widespread and significantly correlates with disruptions in sexual development and decreased fertility. Further research is needed to deepen the understanding of these effects and to raise awareness about the health risks associated with mycotoxin exposure.

Moreover, AOH produced by Alternaria species and commonly found on plant-based foodstuffs [59], exhibits endocrine-disrupting properties due to its structural similarity to estrogen. The H295R cells, derived from human adrenocortical carcinoma, demonstrate the significant modulation of steroidogenic gene expression. This includes the up-regulation of key steroidogenic enzymes and receptors, such as CYP1A1, MC2R, HSD3B2, CYP17, CYP21, and CYP11B2, as well as CYP19, while exhibiting down-regulation of NR0B1. A proteomic analysis confirmed these findings, showing alterations in proteins related to steroid biosynthesis and C21-steroid hormone metabolism [60,61].

PAT, produced by *Penicillium expansum*, is primarily associated with fruits and was initially studied for its potential antibiotic properties [62]. However, PAT can elicit human toxic responses, including nausea, vomiting, and genotoxic effects [62]. It is suspected of being an endocrine disruptor. In male rats, high-dose PAT exposure led to increased plasma testosterone and decreased T4 hormone levels, with longer exposure resulting in elevated testosterone and LH levels [63]. PAT also caused low sperm counts and testicular alterations, including edema and fibrosis [64]. In vitro studies using the H295R cell line indicated that PAT could modulate endocrine function by affecting hormone production and nuclear receptor activity [65].

In summary, mycotoxins such as ZEN, AFB1, AOH, and PAT are potent endocrine disruptors that significantly affect reproductive health by modulating hormone production and impacting sexual development. ZEN is particularly associated with premature sexual development in females and fertility issues, while AFB1 is linked to male infertility. AOH and PAT also disrupt hormone synthesis and affect reproductive organ function. Although the molecular mechanisms behind these disruptions are still under investigation, current research trends focus on understanding these mechanisms, improving detection methods, and assessing the long-term health risks of mycotoxin exposure. There is an increasing emphasis on developing strategies to mitigate these effects and enhance public awareness of the potential reproductive health risks posed by mycotoxin exposure.

### 2.1.2. Mycotoxins and Pregnancy Outcomes

Mycotoxins, such as aflatoxins (AF), ZEN, and others, can adversely affect pregnancy and neonatal outcomes through various mechanisms. ZEN, for example, has been shown to impact placental functions including the maintenance of the placental barrier, cell fusion, and secretion processes. Exposure to ZEN can stimulate cell fusion, leading to the increased secretion of human chorionic gonadotropin (hCG) [66,67]. Additionally, α-zearalanol, a metabolite of ZEN, may increase the risk of preterm labor by elevating the production of corticotropin (CRH) and cyclooxygenase-2 (COX-2) [68]. Furthermore, studies by Partanen et al. have demonstrated the presence of aflatoxin B1 (AFB1) and its metabolites in umbilical cord blood, illustrating the transfer of these toxins across the placental barrier and their dependence on geographic factors [69]. Conversely, Wart et al. highlighted the metabolism of ZEN into estrogenic forms and its rapid transfer across the placental barrier using an ex vivo model [70]. This indicates that the unborn child may be exposed to various mycotoxins and their metabolites, potentially disrupting hormonal balance.

Mycotoxins are recognized as factors that can negatively impact neonatal weight. Research has consistently shown a negative correlation between maternal exposure to AFB1 and birth weight [71–74]. Turner et al. demonstrated that AFB1 exposure during

pregnancy leads to the formation of its metabolite in umbilical cord blood, associated with CYP3A enzyme expression, which suggests adverse effects on neonatal weight [74]. Shuaib et al. also noted reductions in newborn head circumference linked to maternal AFB1 exposure [75]. However, some older studies have not confirmed these associations [76].

Neonatal jaundice has been connected to maternal exposure to AFB1. Abulu et al. reported elevated AFB1 concentrations in cord blood samples from neonates with jaundice, indicating a possible link to maternal mycotoxin exposure [71]. Similarly, Sodeinde et al. found correlations between serum AFB1 levels in newborns and bilirubin levels, suggesting an association with jaundice [77]. Nevertheless, other studies have failed to establish a clear relationship [73,78], highlighting the need for further investigation.

The impact of mycotoxins on miscarriages and stillbirths is less well documented, but some studies have indicated potential risks. Shuaib et al. reported a 35% higher probability of stillbirths with increased maternal AFB1 exposure [79]. Norwegian studies on female farmers exposed to mycotoxins revealed higher incidences of preterm births and late miscarriages, though no definitive link to perinatal deaths was found [80–82]. Another study found a correlation between aflatoxin-contaminated foods and neonatal deaths [83].

Mycotoxins such as AFB1, ZEN, ochratoxins, and fumonisins can also affect fetal development by crossing the placental barrier and causing malformations. Norwegian research identified higher incidences of birth defects like cryptorchidism and hypospadias among children of farmers exposed to these toxins [81]. In contrast, Missmer et al. associated elevated fumonisin levels with an increased risk of neural tube defects and fetal death, based on sphinganine/sphingosine ratios [84].

Preterm birth is another serious outcome linked to mycotoxin exposure, particularly aflatoxins. Some studies have shown a clear association between aflatoxin exposure and preterm delivery, linked to the toxin's effects on maternal cytokines, which may trigger premature cervical ripening and contractions [85,86]. Some studies report a high probability of preterm birth associated with aflatoxin levels [72,85], while others, such as those by Wang et al. and Andrews-Trevino et al., suggest a more nuanced relationship [87,88].

Despite these significant risks, research on mycotoxin exposure in pregnant women and fetuses remains limited. Research on mycotoxin is crucial due to the potential for developmental issues, birth defects, and long-term health effects. However, this research is underexplored, primarily due to limited public awareness of mycotoxin risks and a widespread mistrust of scientific institutions, especially in countries with lower trust in research [89–91]. The lack of education about mycotoxin sources, including contaminated food, environments, and building materials, reduces both public interest and funding for these studies. Building trust and improving public education about mycotoxins could foster greater participation in research, which is essential for protecting vulnerable populations and developing informed policies to prevent exposure [89–91]. Expanding research in this area is critical for better understanding the risks to fetal and maternal health, particularly in regions with high contamination levels.

### 2.1.3. Sex-Based Variations in Mycotoxin Toxicity and Immune Response

Differences in immune responses between the sexes are evident, in the form of higher mortality rates due to infectious diseases in males and a greater prevalence of autoimmune diseases in females. This disparity is partly attributable to sexual hormones. Females exhibit a higher expression of adaptive immune response genes post-puberty, whereas males show an increased expression of innate immunity genes [92]. Additionally, cytokine profiles following lipopolysaccharide-induced inflammation differ between the sexes, with females producing more TNF-$\alpha$ and IL-1$\beta$ [93]. These variations are linked to the differential activity of mitogen-activated protein kinases (MAPKs) in males and females [94,95]. Two mycotoxins that affect MAPK activity and exhibit sex-dependent immunotoxic effects are deoxynivalenol (DON) and T-2 toxin [96,97].

The T-2 toxin, a type A trichothecene produced by *Fusarium* species, is known for its immunotoxicity and cytotoxic effects on the gastrointestinal tract and fetal tissues [98]. Pro-

teomic studies have explored the T-2 toxin's impact in various models. In primary porcine hepatocytes, the T-2 toxin altered lipid metabolism, oxidative stress, and apoptosis, with CYP3A46, a male-dominant cytochrome P450 isoform, implicated in its metabolism [99,100]. However, the sex of the animals was not specified. Similar findings were observed in primary chicken hepatocytes, whereby the T-2 toxin increased mitochondrial mass and ATP in response to oxidative stress [98]. Additionally, proteomic and transcriptomic analyses in female GH3 cells revealed that the T-2 toxin suppressed growth hormone synthesis and altered protein processing, potentially affecting sex-specific metabolic responses [101]. Further research is needed to determine whether these effects are consistent in male models, given the potential differences in hormone regulation between the sexes [102,103].

DON, a prevalent food-associated mycotoxin produced by various *Fusarium* species, targets the intestine and immune system, and has also been implicated in reproductive and teratological effects [22,104]. Studies indicate that males are more sensitive to DON's toxic effects. Males exposed to DON show a reduced feed intake and weight gain, while females exhibit higher levels of IgG, IgA, and CCK [25,105–107]. Males also have higher levels of IL-6 and DON concentrations in organs and plasma, attributed to slower excretion rates [105,107]. However, inconsistent findings from studies on DON excretion in humans highlight the need for larger sample sizes to clarify these differences [108–112].

In conclusion, research highlights significant sex-based differences in immune responses and toxicity to mycotoxins like the T-2 toxin and DON. These variations are influenced by hormonal regulation, with males generally exhibiting an increased sensitivity to certain mycotoxins. Further research is needed to fully understand the mechanisms behind these differences and how they affect human and animal health. This understanding will help in developing more effective risk assessment strategies for mycotoxin exposure.

### 2.1.4. Sex Differences in Cancer Risk Linked to Mycotoxins

Cancer incidence varies between males and females due to a combination of occupational, behavioral, and intrinsic factors, including the influence of sex hormones, sex chromosome-linked genes, and other biological differences [113]. Mycotoxins such as AFB1, OTA, and FB1 have been associated with cancer development in non-reproductive organs. Research has demonstrated sex differences in the cancer risk related to these mycotoxins.

AFB1, produced by *Aspergillus flavus* and *Aspergillus parasiticus*, is a prevalent aflatoxin and is classified as a Group I carcinogen by the International Agency for Research on Cancer (IARC) [114,115]. The liver is the primary site of AFB1 metabolism, where it is converted to aflatoxin-8,9-exo-epoxide by cytochrome P450 enzymes. This exo-epoxide is highly reactive and can interact with DNA, RNA, and proteins, particularly targeting the p53 tumor suppressor gene, thus contributing to carcinogenesis [115,116]. The carcinogenic effect of AFB1 is more noticeable in males compared to females in species including mice, chickens, and humans [117–120]. This sex difference is attributed to variations in metabolism, hormonal influence on liver inflammation, and cancer promotion [120–124]. The androgen receptor and estrogen receptor have opposing effects on hepatocyte proliferation and nucleic acid metabolism through gene transcription [113]. Sex-specific cytochrome P450 enzymes, such as P450-male (2C11) and P450-female (2C12), show different expression levels in rat livers, regulated by hormones [103,125]. Proteomic analyses of hepatocarcinoma patients have highlighted that AFB1 affects pathways involved in detoxification, drug metabolism, antigen processing, and apoptosis [126]. AKR1B10 has been identified as a potential player in AFB1-induced hepatocarcinogenesis, with higher mRNA expression in non-small-cell lung cancer having been observed in men than in women [127]. However, the correlation between AKR1B10 and sex-specific AFB1 effects remains unexplored. Proteomics studies in male mice with type-1 diabetes mellitus exposed to AFB1 showed that the toxin exacerbates diabetes by reducing MUP1, a liver protein predominantly expressed in males [128,129].

OTA, produced by *Aspergillus* and *Penicillium* species, is known for its mutagenic, carcinogenic, immunotoxic, teratogenic, and nephrotoxic effects [130,131]. It has been observed that OTA induces renal tumors more frequently in males than in females [132].

This increased susceptibility in males is linked to differences in metabolism, bioavailability, body weight, and renal transporter expression [133–137]. Proteomics studies using the HEK293 kidney cell line, whose donor sex is unknown, have revealed mitochondrial damage due to OTA. Changes in the mitochondrial proteome, including the depletion of components from complexes II, III, and V, suggest a decrease in ATP levels and increased oxidative stress, leading to apoptosis [138]. ASK1 and Lon Protease 1 (Lonp1) have been identified as important in OTA toxicity. ASK1 mediates oxidative stress-induced apoptosis, and its role may vary by sex [139]. Lonp1 helps maintain mitochondrial function and mitigate oxidative stress, although no sex-specific data are available [140]. In HepG2 liver cells, OTA exposure led to a significant loss of plasma membrane protein content, affecting the membrane structure and function, but the sex of these cells was not considered [141].

FB1, produced by various *Fusarium* species, disrupts sphingolipid metabolism and causes different syndromes in animals. The sex-specific effects of FB1 have been reviewed [142]. In female mice, FB1 induces hepatocellular adenomas and carcinomas, while in male Fischer F344 rats, it causes renal tumors not seen in females [143]. Female mice exhibit heightened immune responses to FB1, whereas male pigs experience more pronounced negative effects on growth, biochemical parameters, and immune responses [144–147]. These observations highlight the need for further proteomics studies that include both sexes to fully understand FB1's toxic effects.

Furthermore, numerous studies have examined the association between carcinogenesis and naturally occurring estrogen disruptors [148–161]. A crucial area of this research focuses on the connection between early exposure to xenoestrogens and the development of chronic diseases, including cancer, later in life [151,152]. ZEN is particularly significant in this context due to its ability to disrupt the endocrine system by affecting gonadal and pituitary functions. This disruption is thought to contribute to the development and progression of breast cancer [150–152]. The strong estrogenic properties of ZEN are also linked to other cancers, such as ovarian, cervical, breast, and prostate cancers. The prolonged consumption of foods contaminated with ZEN may result in serious health risks [153].

In a study by Kuciel-Lisieska et al., it was found that 37% of women with breast cancer had detectable levels of ZEN in their blood. Additionally, higher concentrations of 10.40 ng/mL were reported in patients with benign breast tumors [154]. These findings suggest that ZEN may be a risk factor for breast cancer [154]. Another study assessed the risk of breast cancer associated with exposure to ZEN and its metabolites (α-zearalenol, β-zearalenol, α-zearalanol, β-zearalanol, and zearalanone) by measuring their urinary concentrations in women. This study suggested that α-zearalanol could potentially play a role in the risk of developing breast cancer [155]. Conversely, a different study found no significant differences in the plasma concentrations of ZEN and its metabolites (α-zearalenol and β-zearalenol) between breast cancer and cervical cancer patients and control groups consisting of patients with other diagnoses and healthy female volunteers [156]. The inconsistencies in these findings may be attributed to variations in the methodologies used to measure these mycotoxins [157]. Furthermore, Pajewska et al. analyzed 61 samples, comprising 12 samples with endometrial hyperplasia and 49 with endometrial cancer. They concluded that both ZEN and its metabolites could induce cancer cell proliferation in the uterus and accumulate in uterine tissues [158]. The study by Unicsovics et al. (2024) investigates the potential role of mycotoxins in the pathogenesis of endometrial cancer. The research analyzed the levels of various mycotoxins and their metabolites in blood serum and endometrial tissue samples from 52 participants diagnosed with endometrial cancer, compared to matched controls without a history of endometrial malignancy. The mycotoxins examined included AFs, DON, OTA, T2/HT2 toxins, ZEN, alpha-zearalenol (α-ZOL), and FB1 [159].

The study found significant correlations between higher concentrations of aflatoxins and zearalenone in the presence of endometrial cancer. Notably, higher levels of mycotoxins such as Afs, DON, OTA, T2/HT2 toxins, ZEN, and α-ZOL were detected in endometrial

tissue compared to blood serum. These findings suggest that dietary exposure to these mycotoxins might contribute to the development of endometrial cancer, warranting further research to explore the relationship between mycotoxin exposure and the disease [159].

In summary, mycotoxins such as AFB1, OTA, FB1, and ZEN are associated with cancer risk, with sex differences in susceptibility. Males are more prone to liver and kidney cancers due to differences in enzyme metabolism, particularly for AFB1 and OTA. In contrast, females are more susceptible to endocrine-related cancers, such as breast and uterine cancer, primarily due to the estrogenic effects of ZEN. Studies also suggest that aflatoxins and zearalenone contribute to endometrial cancer risk. These findings highlight the influence of sex hormones and metabolism on cancer development, underscoring the need for further research to better understand these mechanisms.

The evidence for the potential health risks posed by AFB1, FB1, ZEN, OTA, and their metabolites is illustrated by their effects on human health (Figure 2). This illustration depicts the cancer risks associated with four significant mycotoxins—AFB1, FB1, ZEN, and OTA—and highlights the gender-based differences in susceptibility. AFB1 and FB1 are primarily linked to hepatocellular carcinoma and show a higher incidence in males compared to females. ZEN is associated with various hormone-dependent cancers, including breast, cervical, and endometrial cancers, as well as ovarian and prostate cancers. OTA is linked to renal tumors, with males exhibiting a higher predisposition than females. This figure emphasizes the differential impact of mycotoxins on cancer risks based on gender. Table 1 summarizes the physiological responses associated with exposure to different mycotoxins, detailing their impact on reproductive health and other physiological functions.

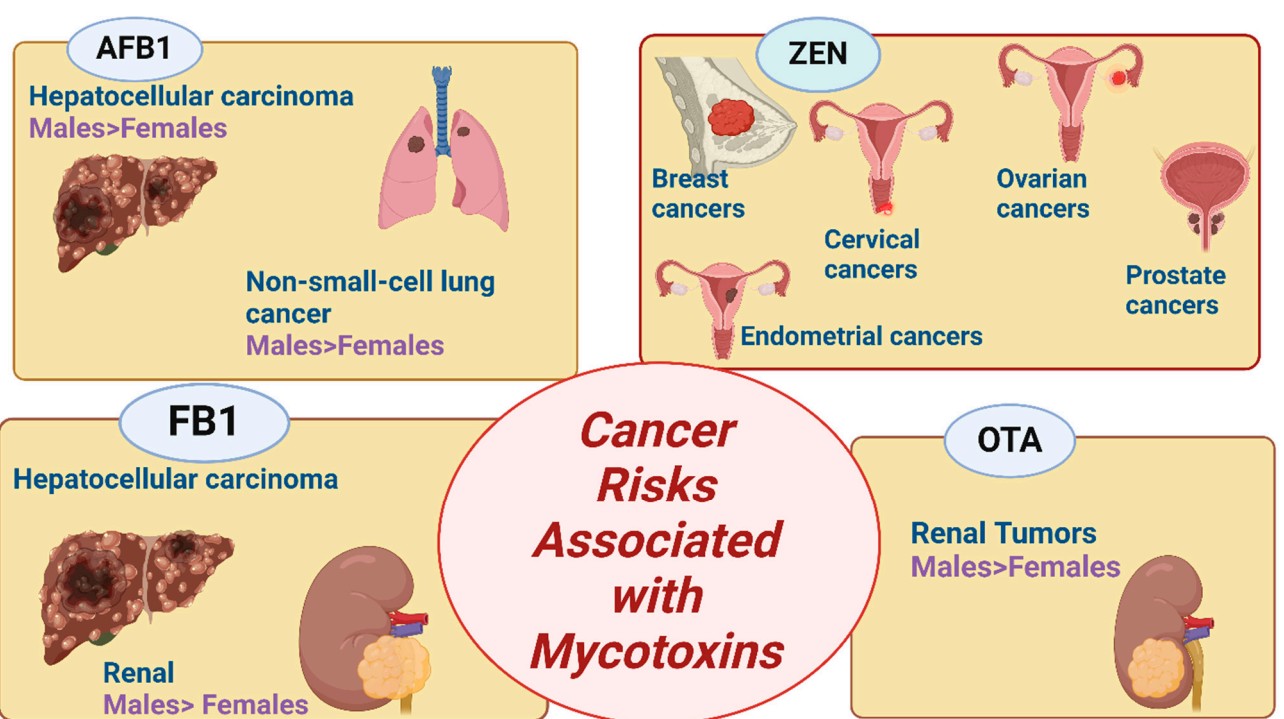

**Figure 2.** Sex-based cancer risks associated with exposure to major mycotoxins: AFB1, FB1, ZEN, and OTA (created with BioRender).

**Table 1.** Physiological responses to various mycotoxins and their effects.

| Mycotoxin | Physiological Response | Details/Effects | References |
|---|---|---|---|
| ZEN | Endocrine disruption in placenta and kidneys | Inhibits HSD11B2, pro-inflammatory effects, and estrogen receptor interaction | [42,43] |
| ZEN | Growth and puberty disorders | Interferes with pituitary and thyroid hormones and leads to growth disorders and delayed puberty | [45] |
| ZEN | Precocious sexual maturation | Higher levels linked to early puberty in girls and boys | [46–49] |
| ZEN | Reproductive issues | Affects sex hormone production (estradiol, progesterone, and testosterone), ovarian folliculogenesis, and can lead to infertility | [52–56] |
| ZEN | Placental function disruption | Affects cell fusion, increases hCG, CRH, and COX-2 secretion, and rapid transfer across placental barrier | [68–70] |
| AFB1 | Male infertility | Found in higher concentrations in infertile men and associated with reduced sperm parameters | [57,58] |
| AF | Reduced birth weight | Negative correlation between AF exposure and birth weight and associated with CYP3A enzyme expression | [71–73,76] |
| AF | Neonatal jaundice | High AFB1 concentrations in jaundiced newborns and correlation with bilirubin levels | [71,73] |
| AF | Miscarriages and stillbirths | Higher probability of stillbirths with maternal AF exposure and correlation with grain farming | [79–83] |
| AF, ZEN, OTA and Fumonisins | Birth defects | Leads to central nervous system malformations, brain damage, and higher incidence of cryptorchidism, hypospadias, and genitourinary defects | [16,81,149] |
| AF, ZEN | Preterm birth | Increases maternal and fetal pro-inflammatory cytokines, leading to preterm contractions and cervical ripening | [72,75,85–88] |
| ZEN, AFB1 | Cancer | Disrupts endocrine system, linked to various cancers (breast, ovarian, cervical, prostate, liver, lung, gastrointestinal, kidney, and gallbladder), and involves mutagenesis and oxidative stress | [150–162] |
| AF | Anemia | Linked to reduced hemoglobin, hematocrit, erythropoiesis, and iron absorption, leading to microcytic anemia | [51,54,85,163–165] |
| Fumonisins | Pre-eclampsia | Associated with increased inflammatory response and higher blood levels in pre-eclamptic pregnancies | [82,95] |

## 2.2. Gender-Based Differences in Dietary Patterns and Their Impact on Mycotoxin Exposure

### 2.2.1. Gender-Based Differences in Dietary Patterns

Eating behavior is influenced by a complex interplay of factors that extend beyond an individual's basic metabolic needs. These factors include food preferences [166], attitudes and beliefs about different food types [167,168], religious practices [169], sociocultural influences [170], and body image perceptions [171,172]. External influences such as social media also play a significant role [173]. Notably, in females, there is a strong association between quality of life, body satisfaction, and a higher likelihood of developing disordered eating patterns [171]. Additionally, energy homeostasis can be disrupted by physiological conditions that impact appetite and caloric requirements, such as metabolic and endocrine disorders [174]. Moreover, climate and environmental conditions play a crucial role in shaping dietary patterns by determining agricultural viability, seasonal food availability, and regional food sources. Climate influences the cultivation of specific crops, such as tropical fruits (bananas, mangoes, etc.) in warm regions [175] or drought-resistant grains (millet, sorghum, etc.) in arid areas, while temperate climates favor grains like wheat and barley [176,177].

Seasonal variations further affect diets, with colder regions relying on preserved foods during winters and tropical areas benefiting from year-round fresh produce [178].

Geographical features, such as the proximity to oceans or forests, also shape diets, with coastal communities consuming seafood-rich meals and inland populations relying on agriculture and livestock. Climate change is disrupting traditional food systems through altered weather patterns, extreme temperatures, and reduced agricultural productivity, particularly affecting subsistence farmers in vulnerable regions. Additionally, growing awareness of the environmental footprint of food production has encouraged shifts toward plant-based diets in many areas. Together, these factors, alongside cultural and economic determinants, highlight the intricate relationship between the climate, environment, and dietary habits [179].

Dietary habits are further influenced by individual likes and preferences, with notable trends emerging between genders. However, research in this area often relies on questionnaires to assess dietary behaviors, which may not always capture the full accuracy of individuals' eating habits [180,181]. Self-reported data may not always accurately reflect actual dietary intake due to factors like recall bias, social desirability bias, and underreporting or overreporting of certain foods. These biases can distort the results, especially when examining gender-based differences in dietary patterns, as individuals may be influenced by societal expectations or personal preferences when responding. Despite these limitations, questionnaire-based studies remain common due to their convenience and cost-effectiveness [180,181].

A recent study by Feraco et al. (2024) utilized an online survey of 2198 participants in Italy to investigate gender differences in food preferences and eating habits [182]. The results indicated that men tend to prefer and consume more red and processed meat, while women are more inclined toward healthier foods such as vegetables, whole grains, tofu, and high-cocoa dark chocolate. Additionally, men are more likely to skip snacks, whereas women tend to eat more frequently, report higher hunger levels in the morning, and experience more frequent episodes of uncontrolled eating without hunger [182].

Similarly, a study in Lebanon by Hoteit et al. (2024) found gendered patterns in food consumption., with a cohort predominantly consisting of females (58.8%) and a mean age of 34.1 ± 12.7 years. The findings revealed significant gender differences in food consumption. Males consumed higher amounts of bread, cereals, grains, dairy products, red meat, processed meat, poultry, fish, and eggs, while females had a greater vegetable intake and a lower consumption of meats and sweets. Males also consumed more water and non-alcoholic beverages, whereas females drank less. Although males slightly consumed more sweets and added sugars, the difference was not statistically significant. These results underscore notable gender-based disparities in dietary patterns and food preferences in the studied population [183].

Women lean more towards healthier diets, deliberate weight control, and eating in social or stressful contexts, whereas men prefer high-fat, strongly flavored meals and eating for pleasure. Men also frequent fast-food restaurants and secretly consume sweets more often than women [162]. Women are more likely to consume whole grains, cereals, vegetables, water, sugar-sweetened beverages, and alcoholic drinks, whereas men consume more eggs, meat, and processed meat, and are more prone to night eating, dining out, and feeling hungrier later in the day [184–186].

Psychological traits add another dimension to dietary patterns. Women tend to prefer healthier options, demonstrate greater conscientiousness, and experience more anxiety, while men exhibit higher extraversion and prefer milk, fermented products, and carbohydrates [166]. Cultural perceptions also shape food preferences, with men consuming more meat often seen as masculine, while fruits and vegetables are considered feminine. Men are generally more resistant to vegetarianism, with traditional gender role conformity predicting a higher consumption of beef and chicken and a lower openness to vegetarianism, while women are more open to vegetarianism for health reasons [163,164].

Meat consumption has strong ties to masculinity, as men show greater defensiveness toward plant-based diets and a more negative attitude toward plant-based eating compared to women [186]. This bias toward meat consumption emerges after age 4, increases with

biological development, and peaks between the ages of 51 and 65 [187]. These findings highlight the multifaceted interplay of gender, psychology, cultural norms, and biological factors in shaping dietary behaviors.

In summary, gender-based differences in dietary patterns are shaped by physiological, psychological, and cultural factors. Men typically consume more red and processed meats, while women prefer healthier foods like vegetables and whole grains. Women tend to focus on weight control and healthier eating, while men prioritize taste and pleasure. Cultural norms associate meat consumption with masculinity, making men less receptive to plant-based diets. Psychological traits also influence eating behaviors, with women being more prone to emotional eating and men displaying higher levels of extraversion. These gendered dietary trends are further influenced by environmental factors, such as the climate and social media, emphasizing the need for gender-specific approaches in nutrition and health interventions.

### 2.2.2. Impact of Gender-Based Dietary Patterns on Mycotoxin Exposure

Several studies underscore gender-specific differences in mycotoxin exposure related to various dietary patterns. Huang et al. (2020) identified higher prevalence rates of AFM1 and FB1 in urine samples from women in the Yangtze River Delta region of China. Conversely, men exhibited higher levels of ZEN, highlighting the necessity for tailored interventions. This study examined the correlations between urinary mycotoxin concentrations and food consumption to identify sources and determinants of mycotoxin exposure. The results indicated a strong positive correlation between urinary AFM1 levels and the consumption of nuts and seeds, consistent with their known contamination by aflatoxins. Additionally, urinary OTA concentrations were significantly positively correlated with the intake of coffee and tea, which are frequently contaminated by OTA. Moderate correlations were observed between AFM1 and the consumption of milk, dairy products, and wheat, with a negative correlation between milk intake and OTA levels, and a positive correlation with FB1 levels. Significant correlations were predominantly found for OTA and beverages, highlighting variability in mycotoxin distribution and dietary intake accuracy [188]. Similarly, a study conducted in Egypt found that adult males consumed sugarcane juice more frequently than females, which potentially increased their mycotoxin exposure risk due to the higher consumption rates [189].

Turner et al. (2010) found that females had slightly higher levels of urinary DON than men, though the difference was not statistically significant ($p > 0.25$). The average DON levels were 6.1 ng/mg (95% CI: 4.6–7.3) in women and 5.8 ng/mg (95% CI: 4.6–7.3) in men. The study also concluded that there was no significant link between gender or other demographic factors and cereal or bread consumption. However, when the researchers adjusted for age, gender, and BMI in a multivariate regression analysis, they found that daily cereal consumption was strongly associated with higher urinary DON levels the next day ($p < 0.001$; adjusted $R^2 = 0.230$) [190].

The study by Turner et al. (2008) investigated the impact of dietary wheat reduction on urinary DON levels in a sample of UK adults [191]. The research involved 25 volunteers who initially followed their regular diet, and then restricted major sources of wheat for a period of four days. Urinary DON levels were measured using a robust assay that included immunoaffinity clean-up and liquid chromatography-mass spectrometry detection. The findings revealed that the mean intake of wheat-based foods decreased significantly from 322 g/day to 26 g/day during the intervention ($p < 0.001$). Correspondingly, urinary DON levels dropped markedly from a geometric mean of 7.2 ng/mg creatinine to 0.6 ng/mg creatinine ($p < 0.001$). This study highlights that dietary modification, specifically reducing wheat intake, can significantly lower urinary DON concentrations [191].

In 2013, a study by Ediage and colleagues examined mycotoxin exposure in 220 children from mycotoxin-contaminated regions in Cameroon. The research found that male children had higher urinary concentrations of DON and FB1 compared to female children, indicating a gender-based disparity in exposure. The weaning status was categorized

into three groups: wholly breastfed, partially breastfed, and fully weaned. Significant differences in aflatoxin M1 concentrations were found across these groups, with the highest levels in partially breastfed children. Dietary habits were assessed, with cassava being the most commonly used staple. Regional differences were also observed, with 53% of households in the Northwest region using maize as the main staple, while 70% used cassava as the primary staple. Peanuts were rarely used as a main ingredient [192].

The study by Hoteit et al. (2024) examined the dietary exposure and risk assessment of multiple mycotoxins (AFB1, AFM1, OTA, OTB, DON, T-2, and HT-2) in the Lebanese food basket consumed by adults, based on the updated Lebanese National Consumption Survey using a total diet study approach [193]. The analysis included 449 participants, with a demographic distribution of 59% females and 41% males. The study reveals that fruits and fruit products are the most commonly consumed items in Lebanese diets, with an average daily intake of 28.19 g. Sugar and confectionery are consumed at a moderate level, while milk, dairy products, cereals, and herbs, spices, condiments, fish, and seafood are consumed at a relatively low level. Beverages, primarily water, are consumed in high quantities, while stimulant and alcoholic beverages are minimally consumed. Age-related differences in food consumption patterns were observed, with older adolescents and young adults having higher intakes of traditional foods, fruit and vegetable juices, and meat products. Older adults showed a higher consumption of legumes and pulses and a lower intake of non-alcoholic beverages. The study also identified key trends in mycotoxin contamination within Lebanese foods, with cereals exhibiting the highest contamination levels [193].

Addressing gender-specific mycotoxin exposure risks request a comprehensive approach that integrates dietary habits, cultural contexts, and biological differences into risk assessment and mitigation strategies. Recognizing that men and women have distinct dietary preferences and behaviors—such as men's higher consumption of red and processed meats and women's inclination towards healthier foods—allows for the development of more precise and effective interventions. Incorporating cultural and psychological factors, such as societal attitudes towards certain foods and gender-based differences in food preparation and consumption, further refines these strategies. By tailoring public health measures to these gender-specific variables, we can enhance the accuracy of risk assessments and the efficacy of prevention and intervention programs. This targeted approach not only reduces the adverse health impacts of mycotoxins but also promotes equitable health outcomes across different populations.

Studies highlight gender differences in mycotoxin exposure due to varying dietary patterns. Women tend to have higher urinary levels of AFM1 and FB1, while men show higher levels of ZEN. Key sources of exposure include nuts and seeds for AFM1, coffee and tea for OTA, and cereals for DON. Gender-based differences in dietary habits, such as men's higher consumption of red meat and women's preference for healthier foods, influence these exposures. Research also indicates that regional and cultural factors affect mycotoxin levels, emphasizing the need for gender-specific risk assessments and tailored public health interventions to improve health outcomes.

### 2.2.3. Occupational Exposure to Mycotoxins

Occupational exposure to mycotoxins primarily occurs through inhalation and dermal contact. While most mycotoxins are not volatile, they can be present in airborne dust, fungal spores, and hyphal fragments [194–197]. These particles can serve as carriers for mycotoxins, which may then be inhaled into the lungs, particularly in environments where airborne dust is prevalent [195,198–201].

In addition to inhalation, dermal contact represents a significant route of exposure, especially in settings in which workers handle contaminated materials, such as food, without adequate protective measures. This risk is heightened in environments in which short-sleeved clothing is worn or hands come into direct contact with mycotoxin-containing solutions [201–203]. Furthermore, mycotoxin-laden dust particles can settle on the skin,

facilitating dermal absorption. Contaminated work surfaces also pose a risk, as they can lead to additional skin contact through touch [202].

Studies on air and surface metrology provide evidence of mycotoxin exposure in environments where organic dust is prevalent. Activities involving a high exposure to such dust, including storage, loading, handling, or milling of contaminated materials (e.g., grain, waste, and feed), pose significant risks. Animal husbandry settings, in which workers manage feed and care for animals, are also notable for elevated exposure risks. In particular, animal feed processing plants are associated with higher mycotoxin exposure due to regulatory limits that permit concentrations of mycotoxins in animal feed to be up to ten times greater than those allowed in human food. For instance, the maximum allowable concentration of DON in unprocessed maize is 1750 μg/kg, compared to 750 μg/kg in cereals intended for human consumption [204].

Biomonitoring research provides crucial insights into occupational mycotoxin exposure, revealing significant differences based on the occupational setting and location. For instance, a study by Malik et al. (2014) in India found elevated levels of aflatoxins in the serum of food-grain workers compared to a control group, suggesting a substantial occupational exposure [205]. Similarly, Saad-Hussein et al. (2014) observed higher serum aflatoxin concentrations among millers and bakers in Egypt, relative to controls [206]. In Portugal, Viegas et al. (2016) detected AFB1 in 50% of poultry workers' serum, whereas it was absent in controls [201]. Conversely, a study by Föllmann et al. (2016) in Germany did not find significant differences in urine spot samples between mill workers and a control group, indicating potential limitations in detecting exposure through urinary biomarkers alone [207]. Further research has expanded to explore dermal exposure; Taevernier et al. (2016) demonstrated that beauvericin and enniatins can penetrate both intact and damaged skin using an in vitro Franz diffusion cell setup, highlighting an additional route of occupational exposure [208].

A summary of 15 biomonitoring studies up to 2015 reveals a predominance of aflatoxins and ochratoxins in initial studies. However, with advances in analytical techniques, more recent studies have addressed multiple mycotoxins in single biological samples [207,209,210]. These studies frequently identify co-exposure to various mycotoxins, illustrating the complexity of exposure assessment. A noted limitation of biomonitoring is its inability to distinguish between exposure from occupational sources versus dietary intake. To mitigate this, many studies included control groups from non-exposed sectors, enhancing the ability to differentiate between workplace exposure and dietary sources [211].

Assessing exposure levels involves understanding the variations between different tasks within the same industry. For example, Viegas et al. (2013b) reported that workers engaged in animal feeding in swine husbandry faced a higher AFB1 exposure compared to those performing other tasks [211]. In waste management settings, a high AFB1 exposure was attributed to waste material contamination, with the exposure levels remaining stable throughout the work shift due to consistent task conditions [212].

The quality of materials handled can also influence the exposure levels. In food processing plants, the contamination levels can vary between batches. Regular monitoring of material contamination and implementing preventive measures are crucial to managing exposure. Contaminated materials should be discarded, and if necessary, personal protective equipment (PPE), such as respiratory masks, gloves, and goggles, should be used. Even low concentrations of mycotoxins in products can result in significant airborne contamination during handling, particularly when dealing with dry materials with high surface areas, such as hay or plant fibers. Manual sorting or transporting contaminated products can release dust and increase exposure levels. Additionally, common tasks, such as cleaning with sweeping or compressed air, are known to cause high dust exposure [213].

Co-exposure to multiple mycotoxins is a common phenomenon, as various mycotoxins often contaminate foodstuffs simultaneously [13,214–217]. Risk assessments should, therefore, account for potential synergistic or additive effects of co-exposure and include measurements for several mycotoxins. Proximity of the worker's head to the material han-

dled also increases exposure risk, underscoring the importance of identifying high-exposure tasks and ensuring proper use of PPE [201,213].

Individuals residing or working in water-damaged buildings, cereal storage facilities, farms, composting plants, or modern office buildings equipped with Heating, Ventilation, and Air Conditioning (HVAC) systems may be exposed to mycotoxins via inhalation [218–220]. This exposure can result from hidden mold growth in indoor environments (e.g., on wallpaper, gypsum boards, carpets, or HVAC systems) or from handling mycotoxin-contaminated food, feed, or waste. The potential for airborne mycotoxin exposure is influenced by factors such as substrate, growth conditions (e.g., temperature, pH, and water activity), and the properties of the fungal species involved [218–220].

The study by Al-Matawah et al. found significant fungal contamination in HVAC systems of a multi-story building in Kuwait, where extreme heat leads to indoor living. High airborne fungal levels, especially of allergenic species like *Aspergillus* spp., were observed year-round. The study indicated that HVAC systems were key in dispersing fungal spores within the building, with higher indoor fungal concentrations than outdoors, particularly when filtration efficiency was lower in colder months. These results emphasize the need for regular HVAC maintenance and filter cleaning to reduce fungal exposure and improve indoor air quality [221].

Moreover, the study by Gołofit-Szymczak et al. in 2023 examined the role of automobile air conditioning (AC) filters as potential sources of fungal contamination and associated health risks, particularly for professional drivers. Their study highlights that over time, AC filters can accumulate mycotoxins produced by fungal species, such as *Aspergillus*, *Penicillium*, and *Fusarium*, which can be aerosolized into the vehicle's cabin. The researchers conducted a seasonal assessment (summer and winter) of fungal contamination in AC filters from 30 randomly selected passenger cars, employing both culture-based and molecular techniques to detect mycotoxins [222].

The study found that fungal contamination was more pronounced in summer, with concentrations averaging $5.4 \times 10^4$ cfu/m$^2$ compared to $2.4 \times 10^4$ cfu/m$^2$ in winter. Notably, *Aspergillus* species, including *A. fumigatus* and *A. flavus*, were frequently identified. These fungi are known for their allergenic and toxic properties, particularly related to respiratory issues and mycotoxin production, such as *aflatoxins*, *ochratoxins*, and *fumonisins* [222].

The study underscores the risk of chronic exposure to airborne mycotoxins, especially for drivers and passengers in confined spaces. It emphasizes the importance of regular maintenance and filter replacement to mitigate health risks from fungal contamination in vehicle AC systems. The findings highlight the need for preventive measures to reduce the exposure of individuals, particularly those with weakened immune systems, to potentially harmful bioaerosols in cars [222].

Similarly, a study by Farian and Wojcik-Fatla (2024) evaluated the effectiveness of cabin filters in passenger vehicles in retaining mycological contaminants, with a focus on fungal contamination and mycotoxin presence [223]. The analysis of 100 cabin filters found an average fungal concentration of $7.2 \times 10^7$ CFU/m$^2$, with *Cladosporium*, *Alternaria*, *Penicillium*, and *Aspergillus* species predominating. The highest concentration was observed in *A. fumigatus*, which also harbored aflatoxins (B1, B2, G1, G2) and ochratoxin A. Carbon filters were found to be more efficient than standard filters in trapping fungal spores and reducing mycotoxin levels. This study suggests that in occupational settings involving prolonged exposure to vehicle air systems, such as for transport workers and delivery drivers, cabin filters with activated carbon should be used and replaced annually to mitigate respiratory risks associated with mycotoxins [223].

These studies collectively emphasize the significance of HVAC and AC systems in the dispersion and accumulation of fungal contaminants and mycotoxins, highlighting the need for regular maintenance and filter replacement to reduce the associated health risks. Preventive measures, such as the use of carbon filters in vehicles and thorough HVAC system maintenance, are essential to mitigate exposure, especially for individuals with weakened immune systems or those spending prolonged periods in confined spaces.

In summary, occupational exposure to mycotoxins primarily occurs through the inhalation of airborne dust, fungal spores, and dermal contact with contaminated materials. High-risk industries such as animal feed processing, grain storage, and food handling show elevated mycotoxin levels, with workers in these settings facing significant exposure, particularly to aflatoxins. Co-exposure to multiple mycotoxins is commonly observed, further complicating risk assessments.

Emerging studies have expanded the focus to indoor environments, particularly HVAC and vehicle air conditioning systems, which can disperse fungal spores and mycotoxins into the air. Research indicates significant contamination in filters, with regular maintenance and the use of activated carbon filters being effective in reducing exposure. This highlights the growing need for comprehensive occupational safety measures, including preventive actions to reduce mycotoxin exposure across different settings, particularly for at-risk individuals. The trend suggests a shift toward considering air and surface contamination, alongside traditional exposure routes, as critical factors in managing occupational health risks associated with mycotoxins.

### 2.3. Gender Differences in Occupational Exposure to Mycotoxins

Although studies on gender-specific occupational mycotoxin exposure are limited, understanding these differences is crucial due to the varied exposure patterns resulting from distinct roles performed by men and women across various industries. Occupational exposure to mycotoxins varies significantly between genders, influenced by the specific tasks and responsibilities each typically undertakes.

In agriculture, women often engage in manual tasks such as sorting, cleaning, and handling contaminated grains, peanuts, and maize. These activities increase their risk of exposure through both dermal absorption and the inhalation of airborne dust. Women are particularly vulnerable to mycotoxin exposure due to inadequate protective measures, such as insufficient use of gloves or long-sleeved clothing, which can result in direct skin contact or the inhalation of dust particles [201–203]. In contrast, men in agriculture are more likely to operate machinery and manage large-scale processing tasks. Their exposure is primarily through the inhalation of dust from bulk materials and handling contaminated substances [211,212].

In the food processing industry, women frequently occupy roles in cleaning, packaging, and quality control, where they handle contaminated ingredients and often lack adequate personal protective equipment (PPE). This close contact with contaminated materials increases their risk of both dermal and inhalation exposure, potentially leading to health issues such as carcinogenicity and endocrine disruption. Men, on the other hand, are more involved in operating machinery and bulk handling, leading primarily to the inhalation of dust from large volumes of contaminated materials. This exposure is associated with significant respiratory risks and potential systemic toxicity.

In animal husbandry, women typically perform routine care tasks, including feeding and cleaning, which exposes them to mycotoxins from contaminated feed and bedding. This exposure can result in acute toxic effects and long-term health complications. Men, however, are more likely to handle feed management and equipment maintenance, leading to the increased inhalation of dust from large-scale feed operations and associated respiratory disorders.

In waste management and recycling sectors, women may engage in sorting and manual handling of waste materials, exposing them to mycotoxins through both direct contact and the inhalation of dust. Men, who often operate heavy machinery and handle larger volumes of waste, face higher risks of inhalation exposure and associated respiratory problems.

Gender-based differences in occupational mycotoxin exposure underscore the necessity for gender-specific occupational health strategies. To effectively mitigate exposure risks and safeguard worker health, it is crucial to implement safety protocols and personal protective equipment (PPE) tailored to these differences. Enhanced safety measures, including the

appropriate use of PPE and routine monitoring of exposure levels, are vital in addressing the distinct risks encountered by both men and women in various occupational settings.

Incorporating these considerations into screening practices and mitigation strategies can significantly improve occupational health outcomes. By focusing on gender-specific needs and refining safety protocols accordingly, workplaces can better protect all employees from the adverse effects of mycotoxin exposure.

*2.4. Gender-Specific Roles and Their Impact on Household Management of Mycotoxin Exposure*

Effective management of mycotoxin exposure in households is intricately linked to implementing preventive measures, as outlined by various guidelines and regulations. Regulatory frameworks aim to safeguard consumer health by setting maximum allowable concentrations and tolerable daily intake values for mycotoxins [224–226]. These measures are accompanied by strategies to control contamination risks at different stages of the food chain [227–229]. However, the household environment plays a crucial role in managing exposure, with gender-specific responsibilities significantly influencing the effectiveness of these measures.

Traditionally, women have shouldered most domestic tasks related to food management, including purchasing, storage, preparation, and cleanliness. Their responsibilities significantly impact adherence to preventive guidelines aimed at reducing mycotoxin risks. The German Federal Institute for Risk Assessment has established seventeen "golden rules" for minimizing mycotoxin exposure, emphasizing buying fresh food, proper storage in cool and dry conditions, and the immediate disposal of moldy items [230]. Similarly, the United States Department of Agriculture (USDA) recommends managing mold presence in food [231]. These guidelines, which include specific recommendations such as cleaning bread boxes regularly, disposing of moldy food, and properly storing cereals and flour, highlight the importance of adequate food management [230]. These practices are particularly relevant to women, who handle most routine food management tasks. Ensuring adherence to these guidelines is crucial for reducing contamination risks. [232,233].

Effective mold management in households requires the implementation of several vital practices. The regular cleaning of refrigerators with baking soda and bleach is essential for removing residual mold and preventing its proliferation. Maintaining the cleanliness of dishcloths and other food-handling items is crucial, as these can harbor mold spores if not properly sanitized. Additionally, controlling indoor humidity levels below 40% helps inhibit mold growth [230,231].

In households in which women typically manage food-related tasks, inspecting produce for mold and avoiding purchasing or consuming moldy items is vital [230,231]. Proper heat processing of homemade preserves is also necessary to destroy potential mold spores. Food protection strategies include covering items to prevent exposure to airborne mold spores, promptly refrigerating opened perishables to minimize mold development, and consuming leftovers within 3 to 4 days to prevent extended mold growth. Adhering to these practices is critical for reducing mold contamination and maintaining food safety [230,231].

Men's roles, such as shopping and handling groceries, are equally crucial in influencing food safety outcomes. In many households, men participate in tasks that significantly affect the quality and safety of food [234]. The effective management of mycotoxin exposure necessitates that both genders are equally informed about food safety practices and involved in their implementation.

Emerging practices like food sharing and dumpster diving, driven by concerns about food waste and anti-consumption, introduce additional risks [235–237]. These informal activities often bypass standard food safety regulations, increasing the likelihood of mycotoxin exposure. Men, who may more frequently engage in dumpster diving for practical reasons, and individuals involved in food sharing, need targeted education on safe food handling to mitigate these risks [235–239].

Gender-specific roles significantly influence the management of mycotoxin exposure in households. Women's traditional responsibilities in food storage and cleanliness are crucial

for implementing preventive measures, while men's roles in purchasing and handling food also affect safety outcomes. To enhance food safety and reduce mycotoxin risks, it is essential to involve both genders in food management practices. Providing comprehensive education on mycotoxin risks and prevention strategies is equally important. Addressing gender-specific roles and offering targeted guidance can improve household food safety and effectively manage mycotoxin exposure.

In conclusion, gender-specific roles shape the management of mycotoxin exposure in households. To improve food safety and reduce risks, it is essential to involve both men and women in food management practices and provide targeted education on mycotoxin prevention.

### 3. Techniques for Mycotoxin Screening

The growing concern over mycotoxin contamination in food products has led to significant advancements in analytical methods aimed at enabling the rapid, sensitive, and accurate detection of these toxic compounds [240]. Among the various methodologies employed, chromatographic techniques are widely recognized for their effectiveness in the quantitative determination of mycotoxins [241,242]. These methods, which include thin-layer chromatography (TLC), liquid chromatography (LC), and gas chromatography (GC), coupled with detection systems such as ultraviolet (UV), fluorescence (FLD), and mass spectrometry (MS), leverage sophisticated instrumentation and comprehensive sample preparation to achieve high sensitivity and precision in mycotoxin detection [241,242].

TLC is one of the earliest established techniques in mycotoxin analysis and is particularly valued for its cost-effectiveness and simplicity in screening multiple samples simultaneously [243]. TLC involves a stationary phase coated on an inert matrix, followed by a mobile phase containing solvents, allowing analytes to migrate through, revealing fluorescent spots under UV light [244]. TLC is effective for qualitative and rapid screening [245], but its quantitative application is limited due to sensitivity and accuracy challenges due to sample preparation complexity and mycotoxins' properties [246].

To overcome the limitations of TLC limitations, LC, particularly high-performance LC (HPLC), is widely used to analyze high-polarity, non-volatile, and thermally labile mycotoxins. HPLC, coupled with fluorescence or UV-visible detectors, allows the direct detection of naturally fluorescent mycotoxins like AFs and OTA, while non-fluorescent ones like FB1 require derivatization [241,246–248]. LC–MS/MS further enhances sensitivity, selectivity, and reliability, enabling multi-mycotoxin detection at trace levels across various matrices [249–251]. GC is useful for volatile mycotoxins, such as trichothecenes and patulin, but requires derivatization due to low volatility [241,242]. Immunochemical methods, particularly enzyme-linked immunosorbent assay (ELISA) and lateral flow immunoassay (LFIA), are commonly employed for large-scale mycotoxin screening due to their simplicity, cost-effectiveness, and sensitivity, although ELISA may face issues with cross-reactivity and matrix validation [243–257].

The evolution of rapid, portable, and user-friendly detection systems, such as LFIA, has substantially advanced the visual detection and semi-quantification of mycotoxins. LFIA systems, known for their reliability, are designed with accessibility in mind, catering to a broad spectrum of users. Commercial kits are now available for the detection of various mycotoxins, including OTA, ZEN, DON, and AFs [244]. The integration of nanomaterials, such as gold nanoparticles (AuNPs), fluorescent nanoparticles (FNs), magnetic nanoparticles, carbon nanoparticles (CNPs), and carbon nanotubes (CNTs), has significantly enhanced colorimetric contrast and detection sensitivity. Nanomaterials improve the optical properties of the assays, leading to more distinct and reliable signal outputs [258–261]. Techniques like enzymatic reactions, such as those catalyzed by horseradish peroxidase, and the use of nanozymes, further amplify sensitivity and assay stability [262–264]. Advanced detection technologies, including fluorescence, surface-enhanced Raman scattering (SERS), and smartphone-based platforms, allow for precise signal detection and quantification, including signals that are otherwise invisible to the naked eye [265,266]. These

technologies, with their precision, enable more accurate and detailed analysis by improving signal resolution and reducing detection limits. Enhancements in transport and reaction kinetics—such as optimized convection, diffusion, and reaction rates—contribute to increased assay sensitivity. Strategies for improving performance include implementing sequential flow, concentrating reactants, and increasing the number of binding sites to boost signal intensity [267]. Furthermore, reducing nonspecific binding is critical for assay accuracy. This is achieved through the optimization of assay components, surface modifications, adjustments in label sizes and concentrations, and the selection of suitable buffer compositions [267]. These advancements in LFIA technology collectively contribute to the development of more precise, efficient, and reliable mycotoxin detection methods, facilitating enhanced food safety.

Recent advancements in mycotoxin analysis and detection have introduced a variety of innovative technologies, each offering unique capabilities and addressing specific challenges in the field. Biosensors, integrating biological elements with transducers, utilize optical (e.g., fluorescence and surface plasmon resonance—SPR), electrochemical (e.g., potentiometric, amperometric, and impedimetric), and piezoelectric (e.g., quartz crystal microbalance—QCM) sensors [268]. These devices often employ advanced nanomaterials such as quantum dots (QDs), metal nanoparticles, nanofibers, and carbon nanotubes (CNTs) to enhance sensitivity due to their high surface-area-to-volume ratio and unique physicochemical properties [269,270].

The electronic nose (e-nose) represents another significant advancement, using an array of chemical sensors to detect volatile organic compounds (VOCs) and identify specific odor profiles associated with mycotoxin-producing fungi [271]. This technology has been effective in detecting OTA in dry-cured pork, aflatoxins (AFs) and fumonisins in maize, and DON in wheat bran [272–275]. However, its application is limited by the need for optimization to detect low concentrations and the challenge of analyzing non-volatile mycotoxins [252].

Fluorescent polarization (FP) immunoassay operates on the principle of competitive binding between a fluorophore-labeled tracer and the target analyte for antibody sites. This method eliminates the need for extensive pre-analytical processing steps required by traditional methods like ELISA, offering increased efficiency [276]. FP has been successfully used to detect various mycotoxins, including ZEN in corn, DON in wheat products, AFB1 in maize, and OTA in rice [276–280]. Despite its efficiency, FP may suffer from lower accuracy and sensitivity compared to high-performance liquid chromatography (HPLC), mainly due to potential cross-reactivity with matrix components and other metabolites [252].

Aggregation-induced emission (AIE) is a novel phenomenon where fluorescent dyes exhibit significantly enhanced emission when aggregated compared to their dilute state [281]. Dyes such as 9,10-distyryllanthracene (DSA), silacyclopentadiene (silole), and tetraphenylethene (TPE) show notable fluorescence in aggregate form [282]. AIE-based aptasensors have been developed to detect OTA in wine and coffee, capitalizing on the high fluorescence of these dyes in aggregated states [280,283].

Emerging technologies, including biosensors, electronic noses, and advanced spectroscopic techniques, provide innovative approaches that enhance sensitivity, specificity, and usability, pushing the boundaries of traditional mycotoxin detection methods. Each of these technologies presents distinct advantages and limitations, requiring careful consideration based on factors like the sample type, detection limits, and cost-effectiveness. As these methods continue to evolve and integrate, they will collectively strengthen the ability to monitor and ensure food safety, protecting public health from the risks associated with mycotoxin contamination in food products.

Advancements in mycotoxin detection have improved sensitivity and speed through methods like thin-layer chromatography (TLC), liquid chromatography (LC), and gas chromatography (GC), with techniques such as LC–MS/MS offering enhanced precision for trace-level detection. Immunochemical methods like ELISA and lateral flowimmunoassays (LFIA) have become popular for their rapid and portable capabilities, further enhanced

by nanomaterials for better sensitivity. Emerging technologies, including biosensors, electronic noses, and fluorescence polarization assays, are pushing the boundaries of detection with greater specificity and ease of use. The trends show a shift towards miniaturized, user-friendly systems, with nanotechnology and biosensors driving innovation, though challenges like cross-reactivity and matrix effects remain. These advancements are enhancing food safety by providing more efficient, accurate, and accessible detection methods.

*Recent Advances in Point-of-Use Mycotoxin Detection*

Point-of-use mycotoxin detection refers to the development of portable, rapid, and user-friendly diagnostic systems that enable the detection of mycotoxins directly at the location of the sample collection [284,285]. These technologies are crucial for real-time monitoring in environments such as food processing facilities, agricultural sites, and consumer markets, where immediate results are needed to assess and mitigate the risks associated with mycotoxin contamination [284,285]. By bypassing the need for complex laboratory procedures, point-of-use detection systems provide a cost-effective and efficient solution for ensuring food safety. Advanced technologies, including LFIAs, optical and electrochemical sensors, and smartphone-based platforms, have significantly enhanced the sensitivity, specificity, and accessibility of these systems, making them valuable tools for improving public health and food security [284,285].

Recent advancements in point-of-use devices for mycotoxin detection have significantly improved in situ assessments, essential for establishing exposure risks in workplaces, food processing facilities, and agricultural sites [286]. The increasing prevalence of mycotoxins in food matrices and their potential health impacts underscore the urgent need for effective and immediate detection methods, primarily focused on enhancing accessibility, efficiency, and reliability by integrating emerging technologies [286].

Innovations in sample interfaces and assay architectures are making mycotoxin detection more portable and user-friendly across various settings [286–288]. Key advancements in sensing surfaces and molecular recognition mechanisms are crucial for enhancing the sensitivity and specificity of detection technologies [289,290]. This involves optimizing detection surfaces and selecting effective molecular mechanisms for recognizing specific mycotoxins. Advanced materials, such as nanostructures and molecularly imprinted polymers (MIPs), offer high specificity for target analytes, which can be integrated into point-of-use devices [291].

Effective in situ sample preparation techniques are critical for real-time monitoring, particularly in environments where exposure risks are high [292]. Simplifying extraction methods helps to reduce matrix effects, ensuring that complex food matrices can be converted into sensor-compatible formats with minimal intervention. The growing demand for point-of-use detection drives the development of techniques that minimize preparation time and resource consumption while maintaining analytical accuracy and sensitivity [293,294].

Recent innovations in sample preparation for complex matrices focus on the rapid isolation of mycotoxins while enhancing selectivity. For liquid matrices like wine and beer, methods such as dilution and matrix neutralization, often combined with selective binding agents like immunoaffinity columns (IACs), effectively concentrate mycotoxins and reduce interferences [291,295]. In solid matrices, solubilization with organic solvents such as methanol and acetonitrile remains standard, but alternative, safer solvents and methods are being explored to enhance operational efficiency [296–309].

Emerging techniques like dispersive liquid–liquid microextraction (DLLME) and aqueous two-phase systems (ATPS) provide efficient extraction with minimal use of hazardous solvents, enhancing selectivity and simplifying post-extraction cleanup, making them suitable for field applications [309–311]. Methanol extraction continues to be widely adopted due to its compatibility with various assay architectures, including commercial lateral flow devices [310–312].

Signal generation and transduction are critical for converting detection events into measurable outputs. Methods are categorized into label-based and label-free systems,

both essential for immediate, in situ analysis [293,294]. Label-based systems utilize optical and electrochemical methods to produce quantifiable signals. Optical techniques like fluorescence, colorimetry, and chemiluminescence convert binding events into detectable signals, providing high sensitivity and user-friendly detection [313,314].

Electrochemical methods also offer highly sensitive detection through electroactive substances that produce measurable electrical signals upon binding to mycotoxins [285,315]. In contrast, label-free systems detect mycotoxins without external labels, relying on intrinsic changes in the sample. Techniques such as surface plasmon resonance (SPR) monitor variations in the refractive index upon mycotoxin binding, enabling highly sensitive detection [316].

Recent innovations in signal transduction aim to enhance sensitivity and accessibility for point-of-use applications. The integration of complementary metal-oxide-semiconductor (CMOS) sensors with LED excitation sources has led to portable optical detection systems, ideal for real-time monitoring in critical environments [297,313].

Advancements in assay architectures, particularly in LFIAs, have significantly improved in situ mycotoxin detection capabilities, making them well suited for point-of-use applications [284–286]. Their simplicity, visual results, and ease of integration into portable devices enhance their practicality, although ongoing optimization is necessary to expand their applicability across diverse detection scenarios [288,289].

In addition to LFIAs, ultra-rapid, single-step assays leveraging the intrinsic fluorescence of specific mycotoxins have been developed, though they remain limited to auto-fluorescent mycotoxins [295,296]. Flow cell assays have miniaturized fluidic paths, reducing sample volumes and speeding up assay times, further enhancing their utility in rapid, field-based detection [296–298].

Notable advancements include using smartphone cameras to measure light transmittance in milliliter-scale containers, facilitating point-of-need diagnostics [313]. The integration of CMOS sensors with LED excitation sources for fluorescence measurements, coupled with microcontroller boards for processing, exemplifies the miniaturization of optical detection systems. Compact optical readers and CCD scanners have been developed for measuring colorimetric, fluorescence, and chemiluminescence signals in lateral flow assays [288,316,317].

Overall, these advancements highlight the evolution of point-of-use mycotoxin detection, with innovations in surface engineering, sample preparation, and signal transduction playing pivotal roles in improving the accessibility, sensitivity, and reliability of these technologies.

In summary, while the underpinning technologies are not in themselves sex or gender biased, considerations of accessibility and acceptability need to be addressed. The trend in mycotoxin detection is moving toward smaller, more portable devices with enhanced sensitivity and faster response times. Innovations in sample preparation, signal transduction, and device miniaturization are making these technologies more practical and effective for real-time, on-site monitoring, expanding their potential applications in diverse environments.

## 4. Regulatory Challenges and Gender-Sensitive Strategies in Mycotoxin Management

### 4.1. Regulatory Framework for Mycotoxins: Challenges and the Need for Stricter Enforcement

Regulating mycotoxin levels in food and feed is essential for safeguarding public health and preventing foodborne illnesses. As these toxic substances produced by fungi can contaminate a wide range of food products, the measures taken to regulate their levels play a critical role in ensuring food safety worldwide. However, the approach to regulation varies significantly across regions, with some countries implementing more stringent standards than others. These regulations are built upon rigorous scientific research that assesses the toxicity of mycotoxins, evaluates food safety risks, and estimates exposure levels.

In the European Union (EU), mycotoxin regulation is especially stringent, with the European Food Safety Authority (EFSA) playing a central role in shaping the regulatory

framework [318,319]. EFSA conducts thorough risk assessments and uses the latest scientific evidence to set maximum allowable limits for a variety of mycotoxins in food and feed. This risk-based approach ensures that the permissible limits evolve in line with emerging data. Regulations such as Commission Regulation (EU) 2023/915 set clear limits for contaminants like aflatoxins, ochratoxin A, patulin, deoxynivalenol, and zearalenone [318]. As new scientific information becomes available, these levels are adjusted to reflect the most current understanding of the associated risks. Additionally, Commission Regulation (EC) No 401/2006 outlines standardized procedures for sampling and testing mycotoxin levels in food products across EU member states, ensuring accuracy and consistency in monitoring efforts. The EU also monitors less commonly regulated mycotoxins, such as Alternaria toxins, citrinin, and ergot alkaloids, to protect consumers from potentially harmful exposure through comprehensive monitoring recommendations [318,319].

On the other hand, the U.S. Food and Drug Administration (FDA) has also made notable updates to its regulations concerning mycotoxins [320]. The FDA's efforts focus on regulating the levels of mycotoxins such as aflatoxins, fumonisins, ochratoxin A, and others. In 2024, the FDA set advisory and action levels for various mycotoxins in foods, including fumonisins in corn products, patulin in apple juice, and ochratoxin A in grains like wheat and barley. The FDA has advanced its testing methods, including the use of multi-mycotoxin liquid chromatography-tandem mass spectrometry (LC-MS/MS), allowing for the simultaneous detection of multiple mycotoxins. This enhances the efficiency and accuracy of mycotoxin monitoring [320]. Additionally, the FDA collaborates with other federal agencies, state departments, and international bodies to align regulatory standards and ensure the safety of food products.

Despite the presence of robust regulatory frameworks in regions such as the EU and the United States (U.S.), there remain significant gaps in the global regulation of mycotoxins. These gaps are particularly evident in low- and middle-income countries (LMICs), where mycotoxin contamination is often more prevalent [321,322]. Factors such as inadequate storage facilities, poor agricultural practices, and climatic conditions conducive to fungal growth contribute to this higher prevalence. Unfortunately, many LMICs lack the necessary resources, infrastructure, and technical capacity to establish or enforce comprehensive mycotoxin regulations. As a result, their regulatory frameworks are either weak or, in many cases, non-existent. This lack of regulation significantly increases the risk of exposure to harmful levels of mycotoxins, posing a serious public health threat [321,322].

In LMICs, food monitoring systems are often underdeveloped, with limited access to advanced analytical tools such as liquid chromatography-mass spectrometry (LC-MS) or immunoassays, which are essential for the accurate detection and quantification of mycotoxins. Furthermore, the enforcement of food safety laws is frequently inadequate due to insufficient funding, a lack of trained personnel, and the limited political prioritization of food safety. These challenges result in higher rates of contamination in local food supplies, some of which are exported to international markets where stricter regulations are enforced. This not only undermines global food safety but also has economic repercussions, as contaminated exports may be rejected by importing countries, leading to significant financial losses for LMICs.

In Africa, for instance, despite growing awareness and efforts to improve food safety, significant challenges remain in the regulation of mycotoxins [89,323–326]. Key issues include the insufficient scientific infrastructure, limited access to advanced analytical tools such as liquid chromatography-mass spectrometry (LC-MS), and unreliable power supplies, all of which restrict the reliability and scope of local research. Many African studies rely on collaborations with institutions in developed countries, as local laboratories often lack the resources to conduct independent research with reproducible results. Additionally, brain drain further exacerbates these challenges, with skilled scientists leaving the region due to poor working conditions, a lack of funding, and limited research opportunities. Addressing these gaps requires investment in infrastructure, training programs for cost-effective and rapid detection methods, and improved access to global scientific resources, such as high-

speed internet and technical support systems. These interventions could significantly enhance local capacity for mycotoxin control and food safety enforcement [89,323–326].

Another critical issue is the inconsistency in mycotoxin legislation worldwide. While regions like the EU and the U.S. have established strict maximum permissible limits for certain mycotoxins, these regulations do not cover the full spectrum of mycotoxins. For many mycotoxins, only guidelines or recommendations exist, rather than enforceable laws. This discrepancy leaves significant regulatory gaps, as non-binding recommendations lack the necessary legal weight to compel compliance. Moreover, even where maximum permissible limits are in place, there is often no legislation imposing severe penalties for violations. Without strict enforcement and deterrent penalties, compliance with these regulations remains challenging, particularly in regions with weak governance or limited resources for regulatory oversight.

To address these issues, it is imperative to strengthen international collaboration and harmonize mycotoxin regulations. Global organizations such as the Food and Agriculture Organization (FAO), the World Health Organization (WHO), and the World Trade Organization (WTO) can play pivotal roles in bridging these regulatory gaps. These organizations can provide technical assistance, facilitate knowledge-sharing, and support capacity-building initiatives in LMICs. For example, they could help establish regional centers of excellence for mycotoxin research, provide training on advanced detection methods, and support the development of national food safety action plans.

Scientific research should also underpin the development of more stringent and enforceable legislation. Evidence-based studies can identify the mycotoxins posing the greatest risks to public health and inform the establishment of stricter maximum permissible limits. Furthermore, the introduction of severe legal consequences for exceeding these limits could significantly improve compliance. Such measures may include fines, mandatory recalls of contaminated products, or restrictions on market access for non-compliant producers. These strategies not only protect public health but also promote fairness in international trade by ensuring that all countries adhere to comparable food safety standards.

In conclusion, the disparity in mycotoxin regulations across the globe underscores the urgent need for harmonization and capacity building. By fostering international cooperation, investing in scientific research, and enforcing stricter penalties for non-compliance, the global community can better protect public health and ensure the safety of the global food supply.

*4.2. Integrating Sex and Gender-Sensitive Strategies in Mycotoxin Management*

Reducing mycotoxin exposure requires scientifically rigorous methodologies that recognize and incorporate sex- and gender-specific considerations. Due to distinct biological and social factors, men and women interact with mycotoxins differently. To achieve the comprehensive control of mycotoxin-related health risks, developing gender-inclusive policies, active community involvement, and customized education and outreach programs is essential. Moreover, regulatory frameworks must be restructured to manage mycotoxins effectively by recognizing these differences.

A critical step in managing mycotoxins is creating risk models incorporating sex- and gender-specific experimental data on exposure routes, dietary habits, employment positions, and metabolic differences in toxin processing. Exposure to mycotoxins often varies by gender due to different roles in food production and agricultural work. Understanding these differences enables the development of targeted interventions [327,328].

Regulatory authorities should require mycotoxin screening and control procedures to consider gender disparities. Establishing gender-specific exposure limits based on sex-segregated data would allow more precise assessments of health risks. Screening should account for gender differences in both exposure and health outcomes. Additionally, community-based treatments must be tailored to reflect the specific roles and responsibilities of men and women in agriculture, food processing, and storage. Actively involving

both men and women in community discussions and decision-making about mycotoxin management ensures that diverse perspectives guide interventions and addresses the needs of all stakeholders [327–329].

Education and outreach efforts must be adapted to bridge the knowledge gaps and address the unique needs of different gender groups. Instructional materials should be tailored to suit men's and women's literacy levels, cultural contexts, and informational needs. These initiatives must leverage each group's most effective communication channels, ensuring maximum reach and engagement [327–329].

Overcoming these challenges requires engaging male community leaders to promote the importance of gender-sensitive mycotoxin management and advocate for mutual respect and understanding. Offering incentives, such as free or discounted testing kits, can encourage women to take proactive steps in reducing mycotoxin exposure in their households.

One of the most critical aspects of integrating sex-sensitive strategies is the development of sex-specific biomarkers for mycotoxin detection. Biomarkers are essential for detecting early toxic effects, and due to differences in metabolism and physiology, they may vary between men and women. Identifying these differences facilitates earlier and more accurate exposure detection, leading to timely interventions. This approach also enables the creation of personalized health recommendations, improving the effectiveness of mycotoxin management individually. Additionally, pregnant women or individuals with physiological conditions that influence mycotoxin metabolism may need specific treatments to protect their health and the health of their children. Healthcare strategies can be tailored based on the distinct physiological responses of each sex, leading to more targeted and efficient treatments.

Integrating sex- and gender-sensitive strategies into mycotoxin management allows for a more comprehensive approach to reducing exposure and improving health outcomes. Accounting for biological and social differences will help develop more precise risk models, implement effective regulations, and design targeted interventions that protect both men and women from the dangers of mycotoxin exposure. Public health campaigns that recognize these differences will improve the adherence to safety guidelines and enhance overall community health and resilience against mycotoxin-related risks.

In summary, effective mycotoxin management requires gender-sensitive strategies that consider the biological and social differences between men and women. This includes developing tailored risk models, regulations, and outreach programs that incorporate sex-specific data on exposure and metabolism. Gender-specific biomarkers for early detection and personalized healthcare are essential for timely interventions. By addressing these disparities, mycotoxin management can be more effective, ultimately improving health outcomes and enhancing public health resilience.

**Author Contributions:** Conceptualization, G.T.; methodology, G.T.; validation, G.T.; investigation, B.S., B.J. and F.T.; resources, H.J.B. and F.T.; writing—original draft preparation, G.T.; writing—review and editing, G.T., B.J., B.S., H.J.B. and F.T.; visualization, F.T.; supervision, B.J., B.S. and F.T.; project administration, F.T. All authors have read and agreed to the published version of the manuscript.

**Funding:** This research received no external funding.

**Data Availability Statement:** Not applicable.

**Acknowledgments:** G.T. thanks the support of PhD fellowship of TU Dublin ARISE 2024.

**Conflicts of Interest:** The authors declare no conflicts of interest.

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
