# Peer review of "Sex- and Gender-Specific Considerations in Mycotoxin Screening: Assessing Differential Exposure, Health Impacts, and Mitigation Strategies"

_2036-7481, doi:10.3390/microbiolres15040165_

Round 1
Reviewer 1 Report
Comments and Suggestions for Authors
The paper addresses a very important issue, that is, the body's exposure to mycotoxins. The paper is written in correct language and its meta-analysis is based on a sufficient number of available publications on this research topic.
Although a few suggestions that occurred to me during the analysis of the manuscript would have been worthwhile to be included in the manuscript.
Subsection 2.2.1 - focuses on liking from a psychological aspect. This is obviously quite an important factor, although from the point of view of the manuscript as a whole, in my opinion it could be shortened a bit.
Subsection 2.2.3 - I haven't read any information on exposure to mycotoxins present in air-conditioning systems, especially filters. Studies confirm that people staying in air-conditioned rooms or also using air-conditioning in cars, are at risk of getting mycotoxins into the respiratory system.
I also miss the aspect on regulations, at least a brief opinion of the authors. Currently, the legislation is not consistent around the world. Second, only for a few mycotoxins are there strict maximum permissible doses in individual products. In contrast, most are recommendations or standards. But no legislation imposes severe penalties when these values are exceeded. And only with the introduction, on the basis of scientific research, of drastic legal methods for exceeding maximum permissible limits are we able to protect the public from exposure to mycotoxins.
The authors point out dietary determinants, mentioning culture, religion or economics as factors influencing the type of products consumed. In contrast, I miss climate and environment as some of the main factors influencing the type of diet consumed.
Research on dietary pattern is largely related to the analysis of questionnaires, which do not always reflect reality- The authors of the paper did not pay attention to the problem of their reliability, and thus the impact on the results of the study.
Research on exposure to mycotoxins of the fetus as well as the pregnant mother is extremely important. but, unfortunately, the number of people willing to do this type of research is small. it is related to the education of the public and general public awareness and lack of trust in scientific institutions in some countries- this aspect was also not paid attention to by the authors.
Mycotoxins have an impact on the development of endometriosis in women - the authors only mentioned this case, and it may be worth a little development of this topic.
Author Response
Dear Reviewer,
Thanks for taking your time to review the manuscript. The comments are very helpful. We have addressed each comments. See the detail the attachment. It is in red colour.

Reviewer 2 Report
Comments and Suggestions for Authors
The paper explores differences in the effects of mycotoxins on the sexes, not only in physiological responses, but also in different patterns of occupational exposure. These findings suggest that gender-specific factors need to be considered when assessing mycotoxin risk and developing mitigation strategies in order to more effectively protect public health and improve the accuracy of risk assessments. But there are some concerns should be addressed.
1. No references have been cited in Part 4. In addition, while the number of references cited in full text is high, there are fewer in the last five years.
2. The structure of the paper needs to be further optimized. The logic of the article is not clear. Chapter 3 is only 3 pages in total.
3. As a literature review, each section should have a summary as well as a statement of trends.
4. Lack of elaboration and discussion of the mechanisms that produce the associated toxicity.
5. What is point-of-use mycotoxin detection?
6. In section 2.1, the article focuses on several typical botulinum toxins. How about other botulinum toxins? Do they have gender-specific effects on cancer risk or immune response?
7. How does section 2.2.1 of the article relate to the toxins discussed in the article? The relationship with toxins should be highlighted and other elements should be simplified.
Author Response
Dear Reviewer,
THanks for your comments. We have addressed each comments in green color.
